# Retrospective identification of key activities in Uganda's preparedness measures related to the 2018–2020 EVD outbreak in eastern DRC utilizing a framework evaluation tool

Christina Potter[1,2]*, Lucia Mullen[1,2], Steven Ssendagire[3], Rhoda K. Wanyenze[3], Alex Riolexus Ario[4,5], Doreen Tuhebwe[3], Susan Babirye[3], Rebecca Nuwematsiko[3], Jennifer B. Nuzzo[6]

1 Johns Hopkins Center for Health Security, Baltimore, Maryland, United States of America, 2 Department of Environmental Health & Engineering, Johns Hopkins University Bloomberg School of Public Health, Baltimore, Maryland, United States of America, 3 Makerere University School of Public Health, Kampala, Uganda, 4 Uganda National Institute of Public Health, Kampala, Uganda, 5 Ministry of Health of Uganda, Kampala, Uganda, 6 Department of Epidemiology, School of Public Health, Brown University, Providence, Rhode Island, United States of America

* cpotter7@jhu.edu

**Data Availability Statement:** Data from grey literature used in this study are publicly available –

## Abstract

Uganda has engaged in numerous capacity building activities related to outbreak preparedness over the last two decades and initiated additional just-in-time preparedness activities after the declaration of the 2018–2020 Ebola Virus Disease (EVD) outbreak in eastern Democratic Republic of Congo (DRC). When Uganda faced importation events related to the DRC outbreak in June—August 2019, the country's ability to prevent sustained in-country transmission was attributed to these long-term investments in preparedness. In order to help prepare countries for similar future scenarios, this analysis reviewed evidence from Uganda's response to the June—August 2019 importation events to identify preparedness activities and capacities that may have enabled Uganda to identify and isolate infected individuals or otherwise prevent further transmission. Content from 143 grey literature documents gathered via targeted and systematic searches from June 6, 2019 to October 29, 2019 and six interviews of key informants were utilized to inform a framework evaluation tool developed for this study. A conceptual framework of Uganda's preparedness activities was developed and evaluated against timelines of Uganda's response activities to the June—August 2019 EVD importation events based on the applicability of a preparedness activity to a response activity and the contribution of the said response activity to the prevention or interruption of transmission. Preparedness activities related to coordination, health facility preparation, case referral and management, laboratory testing and specimen transport, logistics and resource mobilization, and safe and dignified burials yielded consistent success across both importation events while point of entry screening was successful in one importation event but not another according to the framework evaluation tool. Countries facing similar threats should consider investing in these preparedness areas. Future analyses should validate and expand on the use of the framework evaluation tool.

see S1 Text for a full list of citations - and are also available from the corresponding author upon reasonable request. Data generated by interviews with key informants were collected on a not-for-attribution basis due to the sensitive nature of interview content and to encourage transparency among interviewees. Due to this reasoning, individual interview notes are not available for review. Please, contact interviewee organizations for more information regarding their response experiences: World Health Organization, the Joint Mobile Emerging Disease Intervention Clinical Capacity Program, the United Nations Children's Fund, the United States Centers for Disease Control and Prevention, Uganda Ministry of Health, and the United Nations High Commissioner for Refugees. For further information regarding ethical restrictions on releasing individual interviewee data, please contact the Institutional Review Board at Johns Hopkins Bloomberg School of Public Health (jhsph.irboffice@jhu.edu) and the Higher Degrees, Research and Ethics Committee at Makerere University School of Public Health (hdrecadmin@musph.ac.ug).

**Funding:** Funding was generously provided by the Open Philanthropy Project (90085575) Recipient: JN. The funders had no role in study design, data collection and analysis, decision to publish, or preparation of the manuscript.

**Competing interests:** The authors declare that they have no competing interests.

## Introduction

There have been more than 20 known outbreaks of Ebola Virus Disease (EVD) in sub-Saharan Africa since 1976. Most recently, the Democratic Republic of Congo (DRC) has been battling concurrent EVD outbreaks for almost two and a half years, including the second most deadly EVD outbreak ever recorded–the 2018–2020 eastern DRC outbreak. The DRC Ministry of Health declared the 2018–2020 eastern DRC outbreak in North Kivu province on August 1, 2018 [1]. By the time the World Health Organization (WHO) declared the eastern DRC outbreak over on June 25, 2020, 3,470 cases and 2,287 deaths had occurred in South Kivu, North Kivu and Ituri provinces with an overall case fatality ratio of 66% [2,3].

Throughout the 2018–2020 eastern DRC outbreak, neighboring countries were considered at risk for EVD importation due to high cross-border movement attributed to migration of refugees, movement of merchants as well as cross-border family connections. While all of these at-risk countries were motivated to conduct various preparedness activities guided by both in-country expertise and assistance by out-of-country partners, Uganda, Burundi, Rwanda and South Sudan were uniquely singled out by the WHO as "Priority 1 countries" for preparedness activities due to their close proximity to reported EVD cases and large-scale border crossing patterns [4]. These preparedness activities were tested when Uganda experienced two clusters of EVD importation events during June—August 2019 that ultimately did not lead to in-country transmission [4–6]. Uganda's ability to prevent in-country transmission garnered praise in the international health community as well as interest in how the country managed the feat [7].

Uganda has had previous experience combating EVD outbreaks in 2000, 2007, 2011 and 2012 [1] in addition to making significant investments in capacity building over the last two decades. These investments in preparedness prior to the 2018–2020 eastern DRC outbreak have included a number of initiatives such as the development of a National Task Force (NTF) dedicated to outbreak preparedness and response composed of multidisciplinary experts from the Ugandan government, academia and non-governmental organizations as well as the creation of similar district level task forces for coordination of targeted, local-level activities. An Emergency Operations Center (EOC) was built to aid in national coordination along with the national adoption of the WHO Integrated Disease Surveillance and Response Strategy. A network of responders trained to combat viral hemorrhagic fever (VHF) outbreaks was organized with the creation of national and district-level rapid response teams as well as a national directory of trained healthcare workers and village health teams. Extensive capacity building of the laboratory system and specimen transport network occurred, including standardization of forms, which aided in a significant reduction in time between case reporting, case confirmation and response initiation. A permanent national isolation facility was prepared for VHF cases in Entebbe [8]. Additionally, a Field Epidemiology Training Program was established and put into practice with over 400 graduates [9]. In addition to the above capacities developed prior to the 2018–2020 eastern DRC outbreak, Uganda also initiated additional preparedness activities after the 2018–2020 eastern DRC outbreak was declared in the North Kivu province prior to the June—August 2019 importation events and resulting activation of response activities [10].

Other reports have asserted that Uganda's long-term capacity building investments and just-in-time preparedness activities cumulatively lead to the country's success in managing the June—August 2019 importation events and preventing in-country transmission [1,11,12]. However, these reports have not systematically evaluated specific preparedness activities, capacities and strategies that were most useful in Uganda's successful containment of EVD. In order to better understand what actions and capacities may be most helpful in preparing for

and responding to EVD outbreaks, we reviewed evidence from Uganda's response to identify specific preparedness activities and capacities that may have enabled Uganda to withstand EVD importation events related to the 2018–2020 EVD outbreak that occurred in neighboring eastern DRC. We reviewed the existing grey literature and conducted interviews of key informants, using a framework evaluation tool developed for this study, to identify which preparedness capacities and activities were found to be most useful in helping to contain imported EVD cases. The goal of this analysis was to inform the development or strengthening of other countries' EVD preparedness plans by identifying highest priority capacities and preparedness activities that may most contribute to the successful containment of Ebola.

## Materials and methods

### Ethics statement

This study received ethical approvals from the Institutional Review Board at Johns Hopkins Bloomberg School of Public Health (#IRB00009609) and the Higher Degrees, Research and Ethics Committee at Makerere University School of Public Health (#721). Clearance was also obtained by the Uganda National Council of Science and Technology. All participants provided informed consent with study information provided in a letter attached to email invitations. Interviewees provided verbal consent at the beginning of their interview when prompted by the interviewer, including consent for audio recording and written notetaking. Verbal consent was witnessed by multiple project team members and documented in interview notes.

Initially, data was collected to inform the framework evaluation tool from relevant open-source grey literature and a selection of key informant interviews. The grey literature search strategy was conducted with both targeted and systematic searches. Websites of organizations and agencies that were well-known actors during Uganda's preparedness and response activities after the declaration of the 2018–2020 eastern DRC outbreak were searched for official report documents that described Uganda's Ebola preparedness and response operations from August 1, 2018 onwards. Websites of the following organizations were included in this targeted search: the Uganda Ministry of Health, WHO, the World Food Programme, the United Stated Centers for Disease Control and Prevention (US CDC), the United Nations High Commissioner for Refugees (UNHCR), Save the Children, the International Federation of Red Cross and Red Crescent Societies, the International Organization for Migration, Médecins Sans Frontières (MSF), the United Nations Children's Fund (UNICEF) and the United States Agency for International Development. A more systematic search for grey literature was also conducted utilizing ReliefWeb (See Fig 1). Results of both search approaches were then screened, excluding any documents that were irrelevant to Uganda's preparedness and response activities. Documents that were duplicative or specific only to non-governmental organization operations and not to Uganda's overall preparedness and response activities were also excluded. Data collection of grey literature occurred from June 6, 2019 to October 29, 2019.

143 grey literature documents in total were included in the creation of the framework analysis. 96 of these documents were WHO Uganda Ebola Virus Preparedness Update reports, a group of highly detailed reports co-published by WHO Uganda and Uganda's Ministry of Health regarding all EVD-related preparedness activities in Uganda spanning from August 1, 2018 to May 30, 2019 [10]. A regional overview report of Ebola preparedness by multiple United Nations agencies was also included [4]. In addition, 32 Ebola Virus Disease Outbreak Uganda Situation Reports, a series of reports co-published by WHO Uganda and Uganda's Ministry of Health regarding response activities to the June 2019 EVD importation in Uganda,

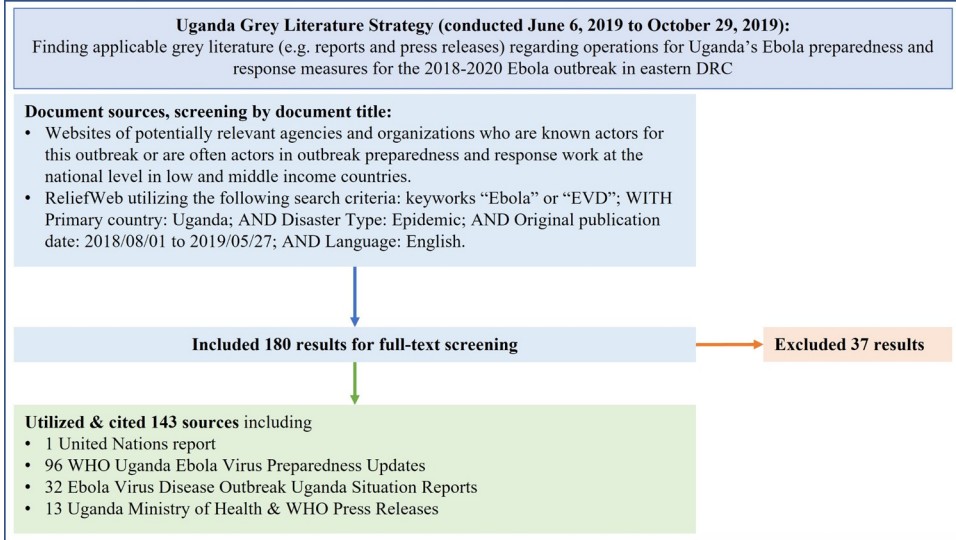

**Uganda Grey Literature Strategy (conducted June 6, 2019 to October 29, 2019):**
Finding applicable grey literature (e.g. reports and press releases) regarding operations for Uganda's Ebola preparedness and response measures for the 2018-2020 Ebola outbreak in eastern DRC

**Document sources, screening by document title:**
• Websites of potentially relevant agencies and organizations who are known actors for this outbreak or are often actors in outbreak preparedness and response work at the national level in low and middle income countries.
• ReliefWeb utilizing the following search criteria: keyworks "Ebola" or "EVD"; WITH Primary country: Uganda; AND Disaster Type: Epidemic; AND Original publication date: 2018/08/01 to 2019/05/27; AND Language: English.

**Included 180 results for full-text screening**                    **Excluded 37 results**

**Utilized & cited 143 sources** including
• 1 United Nations report
• 96 WHO Uganda Ebola Virus Preparedness Updates
• 32 Ebola Virus Disease Outbreak Uganda Situation Reports
• 13 Uganda Ministry of Health & WHO Press Releases

**Fig 1. Grey literature search strategy.**

were also used [5]. Uganda's Ministry of Health and WHO press releases regarding Uganda's response to EVD importations events were utilized as well [6]. See S1 Text for the full references list for the 143 documents included.

A selection of remote key informant interviews to supplement the grey literature collection were conducted with a selection of agencies involved in Uganda's preparedness and response activities during the 2018–2020 eastern DRC outbreak. Semi-structured interviews were conducted on a not-for-attribution basis with public health practitioners and leaders. Interviews were jointly conducted by the Johns Hopkins Center for Health Security and Makerere University School of Public Health. See S2 Text for semi-structured interview guide. While original plans included conducting more national level interviews as well as interviews at the district level for further insight, the COVID-19 pandemic required the authors to pause their activities as the experts selected for interviews became involved in the COVID-19 response and, due to human resource and other constraints, the authors were not able to continue with interviews. A total of six interviews were completed prior to the COVID-19 pandemic with personnel from WHO, the Joint Mobile Emerging Disease Intervention Clinical Capacity Program (JMEDICC), UNICEF, US CDC, Uganda Ministry of Health, and UNHCR.

A novel framework evaluation tool was developed for this analysis, informed by data collected from grey literature and key informant interviews. Initially, a conceptual framework of Uganda's preparedness was developed, identifying the overall scope of preparedness efforts conducted in Uganda after declaration of the 2018–2020 eastern DRC outbreak but prior to the June—August 2019 EVD importation events with preparedness activities grouped by activity type. Activity types were created via thematic analysis of grey literature and key informant interview content by expert researchers. Then, importation event response timelines were developed, identifying the timeline of events and activities that occurred in Uganda during the June—August 2019 EVD importation events and the associated activation of Uganda's response. The preparedness framework activity types were then piloted by expert researchers who appraised the application of preparedness activity types to response activities and events from the importation event response timelines.

After this piloting, the preparedness framework was evaluated against each response timeline for the June—August 2019 EVD importation events by expert researchers based on

thematic analysis of the data collected and previous general knowledge of outbreak preparedness and response operations and capacities. For each response timeline, events and response activities that occurred were appraised by preparedness activity type. Preparedness activity types precipitating response activities that reasonably contributed to the interruption of chains of transmission or prevention of transmission through the protection of susceptible individuals were identified as potential contributors to the prevention of in-country transmission during the June—August 2019 importation events. Preparedness activity types precipitating response activities that were identified as unsuccessful in the interruption of chains of transmission or prevention of further transmission through the protection of susceptible individuals were also classified accordingly. If a preparedness activity type was not involved in building capacity for a particular response activity, it was classified as not applicable for that activity. All classifications were reviewed by all members of the project team. If there was remaining uncertainty regarding the applicability of a preparedness activity type to a response activity or the contribution of a response activity to the prevention of transmission after review, then that uncertainty was also noted accordingly. Overall classifications for each preparedness activity type for each importation event response timeline were then assessed based on previous classifications by response activity. This final importation event response timeline evaluation was reviewed by all project team members as well.

## Inclusivity in global research

Additional information regarding the ethical, cultural, and scientific considerations specific to inclusivity in global research is included in S3 Text.

# Results

## Preparedness framework

Data from grey literature and key informant interviews were organized into eleven preparedness activity types to describe the scope of preparedness activities undertaken in Uganda after the declaration of the 2018–2020 eastern DRC outbreak but prior to the June—August 2019 importation events and the associated activation of Uganda's response. For example, coordination activities formed an activity type, involving activation of the NTF and District Task Forces (DTFs), which would meet regularly throughout the response. The NTF determined the nation's overall strategy, planning and budget and addressed problems and patterns seen across districts. DTFs managed planning and implementation at the district level based on guidance from the NTF and assisted by technical support or resources from partners. Subcommittees on task forces managed subject or action-specific implementation and problem-solving. Subcommittees of these task forces were responsible for conducting risk, needs and readiness assessments at the national and district levels, including the identification of high risk districts bordering DRC whose preparedness would be prioritized as well as the continual reassessment of preparedness activities. Feedback from community members and preparedness-focused workers and officials was considered for changing and improving preparedness activities as well. Formal inspections and assessments, drills and simulation exercises were also utilized regularly. Rapid communication and collaboration with DRC and key partners including foreign and domestic non-governmental organizations, United Nations agencies and other national governments were another key facet of coordination activities along with the creation and use of a rapid and robust communication network between stakeholders. Coordination activities also included the management and deployment of small teams of out-of-country technical support as well as the mobilization and refresher training of the Uganda National Rapid Response Team.

Surveillance and case finding made up another preparedness activity type and included activities such as training of facility and non-facility based health workers, active surveillance at the local and district level, surveillance at cross-border marketplaces, contact tracing and follow-up of alert and suspect cases that later tested negative or were otherwise ruled out, passive surveillance through health facilities, reporting through media platforms and utilization of health information systems. Community engagement, risk communication and social mobilizations activities made up an additional preparedness activity type and were comprised of consistent, transparent communication of government activities via press releases and reports as well as engagement of the public in local languages via multiple platforms including in-person, radio, television, social media, print materials and telephone with a focus on addressing at-risk groups and false rumors. Health facility preparation was another preparedness activity type and included identification and preparation of EVD isolation areas and treatment facilities, vigorous health worker training campaigns, usage of technological platforms to raise awareness of EVD among facilities and the monitoring of relevant staff issues such as complacency or lack of pay.

Case referral and management was a preparedness activity type that included the identification and preparation of EVD isolation and transit facilities, and the creation and preparation of district case management teams. Lab testing and specimen transport was an additional activity type and was focused on improving speed and safety of testing including bolstering of the hub specimen transport system via standardization of forms, training of health workers in sample procurement and prepositioning of transport vehicles. Vaccination activities including vaccinating front-line staff, capacity building for ring vaccination as well as related training formed another activity type. It should be noted that preparation for ring vaccination activities was initially delayed due to delays in protocol approval.

Refugee-related measures formed an activity type, involving health facility preparation, screening, symptom monitoring for newly arrived refugees, surveillance and community engagement activities specific to the vulnerable population and the districts that house them. Logistics and resource mobilization activities formed an activity type and was comprised of prepositioning of critical stockpiles and supplies along with continuous inventory checks and needs assessments. Point of entry screening formed another activity type and was made up of identifying locations for point of entry screening facilities, conducting screening of people entering the country as well as the building, supplying, and staffing for those locations. An additional activity type was safe and dignified burial activities including the formation, training and supplying of burial teams. See Table 1 for a visual summary of the eleven preparedness activity types.

## Importation event response timelines and framework evaluation

Timelines were formed for the two EVD importation events in Uganda connected to the 2018–2020 eastern DRC outbreak and a framework evaluation was conducted across both importation events to identify preparedness activities in Uganda that yielded a likely benefit for response activities during the June—August 2019 importation events.

**Uganda's first importation event.**   In the lead up to Uganda's first importation event in June 2019, family members of a deceased confirmed EVD case in DRC were identified by the DRC government as EVD contacts and began to travel towards Uganda. DRC health officials later identified the travelers as suspect cases and transferred the twelve family members to an Ebola Treatment Unit in Beni. Six of the family members left isolation and crossed the border into Uganda. Point of entry screening did not identify the symptomatic family crossing the border, but DRC authorities notified Ugandan authorities of the crossing and provided

**Table 1. Preparedness framework by activity type.**

| Activity type | Description |
|---|---|
| Coordination | • Activation of and regular meetings for NTF and DTF<br>• Planning, funding and implementation of national strategy<br>• Assessment of national and district needs, readiness and performance<br>• Oversight and coordination of deployment and distribution of technical support and other resources<br>• Rapid coordination and collaboration with partners |
| Surveillance & case finding | • Health worker training<br>• Active surveillance at the local and district level<br>• Contact tracing and follow-up of alert and suspect cases<br>• Passive surveillance through health facilities<br>• Reporting across multiple information and media platforms |
| Community engagement, risk communication & social mobilization | • Regular reporting on country activities<br>• Utilization of multiple media platforms and modes of engagement<br>• Translation into local languages<br>• Usage of targeted messaging to reach at-risk populations and dispel harmful, false rumors<br>• Health worker training |
| Health facility preparation | • Identification and preparation of isolation and treatment facilities<br>• Health worker training and awareness raising<br>• Assessments of facility and staff needs and readiness |
| Case referral & management | • Identification and preparation of temporary isolation and transit facilities<br>• Creation and training of district case management teams |
| Lab testing and specimen transport | • Increased speed of testing via bolstering of transport system, prepositioning of supplies and standardization of protocols<br>• Increased safety of testing via health worker training |
| Vaccination | • Front-line worker vaccination<br>• Ring vaccination preparation<br>• Preparation of vaccination teams |
| Refugee-related measures | • Screening and symptom monitoring of incoming refugees<br>• Enhanced surveillance and community engagement activities in refugee communities<br>• Preparation of health facilities that serve refugee populations |
| Logistics & resource mobilization | • Prepositioning of supplies<br>• Inventory assessments |
| Point of entry screening | • Identification, preparation and maintenance of point of entry screening facilities<br>• Conducting point of entry screening |
| Safe & dignified burials | • Preparation and maintenance of burial teams<br>• Conducting safe and dignified burials |

relevant information regarding the suspect cases. Uganda quickly disseminated this information to health facilities. On June 10, 2019 the family sought care at Kagando Hospital in Kasese district for a child in the party who had developed EVD-associated symptoms. Staff at the hospital quickly isolated the child and transferred him to Bwera Hospital Ebola Treatment Unit. The child's blood sample was sent for testing to UVRI. By June 11, 2019, the other five family members were also placed in isolation at Bwera Hospital Ebola Treatment Unit.

On June 11, 2019, the symptomatic child's lab results came back positive for EVD, becoming the first confirmed case of EVD in Uganda related to the 2018–2020 EVD outbreak in eastern DRC. The Uganda Ministry of Health made a formal public announcement regarding the confirmed case. A rapid response team from the Ministry of Health, WHO and US CDC was dispatched to Kasese. Contact tracing and active case searches were conducted, particularly among health facilities visited by the family with up to 112 contacts line listed and followed up

on by June 14. Psychosocial support teams were deployed, and risk communication activities were amplified. Point of entry screening was put on alert. Sensitization and refresher trainings were conducted for responders and health workers. Vaccination of front-line workers continued, and ring vaccination of contacts began on June 15, although the vaccination of contacts was delayed due to delays in the protocol approval process for use of the vaccine. Logistical assessments were executed to identify and respond to gaps in resources or other needs. Regular task force meetings were conducted with Uganda Ministry of Health, partners, and Kasese District Task Force. Mass gatherings were temporarily suspended.

The five-year old child died on June 11 and received a safe and dignified burial. Two other family members–the 3-year-old brother of the first case and their grandmother—became symptomatic on June 11 with test results sent to Uganda Virus Research Institute coming back positive on June 12. The grandmother died from her illness the night of June 12 and received a safe and dignified burial. Ugandan leaders coordinated with DRC officials to have the ill 3-year-old brother and the rest of the family members along with a Ugandan national married to a member of the family repatriated to DRC for isolation and treatment and to be with the other ill family members isolated in the Beni Ebola Treatment Unit. Uganda and DRC also signed a Memorandum of Understanding to arrange future repatriation of cases as needed for treatment and assuring the continuation of cross-border surveillance and staffing of points of entry on both sides of the border.

Additional suspect cases and alert cases emerged, but all were cleared and no further confirmed cases were reported. No health workers were reported as infected during the course of the response. Logistics and resource mobilization efforts were sufficient enough to allow the response activities above to occur. The ability for all of these components to interact successfully may be attributed coordination activities such as routine simulation exercises and continuous trainings and reassessments of preparedness conducted prior to the importation event as well as strong communication and collaboration relationships between national responders, local responders and Ugandan officials and partners. Additionally, Kasese was identified early in preparedness activities as a high risk district, which may have led to its ability to respond successfully.

Based on the above information, the framework evaluation for the first importation event found coordination, surveillance and case finding, health facility preparation, case referral and management, lab testing and specimen transport, logistics and resource mobilization, and safe and dignified burials as areas of preparedness that played a positive contributing role to the response. Preparedness activities related to community engagement, risk communication and social mobilization, vaccination, and refugee-related measures may or may not have played a positive contributing role in the importation event response. Preparedness activities related to point of entry screening did not play a positive contributing role for the importation event response. See Table 2 for a detailed summary of the framework evaluation for the first importation event.

**Uganda's second importation event.**   On August 28, 2019, a girl and her mother traveled from DRC into Uganda via Mpondwe border crossing in Kasese district in order to seek care at Bwera Hospital. The traveling party cooperated with point of entry officials who correctly identified that the girl in the party had symptoms consistent with EVD infection: fever, fatigue, rash and bleeding from the mouth. The girl was isolated and transferred to Bwera Hospital Ebola Treatment Unit. Specimens were taken and sent to UVRI who confirmed that the girl tested positive for EVD on August 29, 2019. The girl died on August 30, 2019. The deceased remains of the girl were repatriated to DRC. Uganda Ministry of Health dispatched a rapid response team to Kasese for follow-up and district teams continued to conduct community engagement activities, contact tracing, psychosocial support and vaccination activities. Similar

**Table 2. Framework evaluation for first importation event.**

| June 2019—Importation timeline | Coordination | Surveillance & case finding | Community engagement, risk communication & social mobilization | Health facility preparation | Case referral & management | Lab testing & specimen transport | Vaccination | Refugee-related measures | Logistics & resource mobilization | Point of entry screening | Safe & dignified burials |
|---|---|---|---|---|---|---|---|---|---|---|---|
| Border crossing | ? | ? | ? | N/a | N/a | N/a | ? | ? | ? | X | N/a |
| Cases identified and isolated at Kagando Hospital after DRC notified Ugandan officials and health facilities put on alert | ✓ | ✓ | ? | ✓ | N/a | N/a | ? | ? | ✓ | N/a | N/a |
| Cases transferred to Bwera ETU | ✓ | N/a | N/a | ✓ | ✓ | N/a | ? | ? | ✓ | N/a | N/a |
| Isolation and treatment at Bwera ETU | ? | N/a | N/a | ✓ | N/a | N/a | ? | ? | ✓ | N/a | N/a |
| Lab testing of cases | ✓ | N/a | N/a | N/a | N/a | ✓ | ? | ? | ✓ | N/a | N/a |
| Follow up contact tracing and case finding | ? | ? | ? | ? | N/a | ? | N/a | ? | ? | ? | N/a |
| Vaccination activities | ? | N/a | ? | ? | N/a | N/a | ? | ? | ? | N/a | N/a |
| Safe and dignified burials | ✓ | N/a | ? | ? | N/a | N/a | ? | ? | ✓ | N/a | ✓ |
| Repatriation of living cases | ✓ | N/a | N/a | ? | ? | N/a | ? | ? | ✓ | N/a | N/a |
| Event evaluation | ✓ | ✓ | ? | ✓ | ✓ | ✓ | ? | ? | ✓ | X | ✓ |

✓ refers to mechanisms that operated successfully and played a positive contributing role to the response. X refers to mechanisms that operated unsuccessfully or did not play a positive contributing role to the response. ? refers to mechanisms that operated with mixed success or limited information; there was uncertainty as to the positive or negative contributing role to the response. N/a refers to mechanisms that were not applicable to response operations.

to the previous importation event, execution of national simulation exercises, identification of Kasese as a high risk district, and strong communication and collaboration network between local responders, national responders and partners may have also played a positive contributing role. No health workers were reported as infected during the course of the response, and no additional confirmed cases were reported.

Based on the above information, the framework evaluation for the second importation event found coordination, community engagement, risk communication and social mobilization, health facility preparation, case referral and management, lab testing and specimen transport, logistics and resource mobilization, and point of entry screening as areas of preparedness that played a positive contributing role to the response. Preparedness activities related to surveillance and case finding, vaccination, and refugee-related measures may or may not have played a positive contributing role in the importation event response. Preparedness activities related to safe and dignified burials were not applicable to this particular importation event response. See Table 3 for a detailed summary of framework evaluation for the second importation event.

**Overall framework evaluation.**   Preparedness activities related to coordination, health facility preparation, case referral and management, lab testing and specimen transport, logistics and resource mobilization, and safe and dignified burials yielded consistent success across both importation events by interrupting chains of transmission or preventing transmission to susceptible individuals; therefore, it is indicated that these preparedness activities may have strongly contributed to Uganda's success in preventing in-country transmission (see Table 4). Point of entry screening activities uniquely were successful in one importation event but unsuccessful in another, so it seems as though those preparedness activities may or may not have contributed to Uganda's success. Other activities may have also contributed to Uganda's success in preventing sustained transmission in-country but there is a less clear relationship between those activities and prevention of transmission: surveillance and case finding, community engagement, risk communication and social mobilization, vaccination, refugee-related measures.

## Discussion

Using a framework evaluation tool developed for this evaluation, we were able to identify specific preparedness capacities and activities that were referenced as being successful in containing the spread of imported cases of EVD. While there are currently established best practices for national preparedness strategies related to EVD outbreaks, prior to this study there was, to our knowledge, no existing framework to evaluate comprehensive national efforts to prepare and respond to EVD importation events in order to identify in a systematic fashion areas a country's preparedness and response strategies in need of improvement. Additionally, this method can work well in conjunction with after action reports and simulation exercises, which can also provide assessments of preparedness and response activities in response to importation events. In the past, after action reports, simulation exercises and information from other evaluations were reviewed individually, sometimes inconsistently. Conversely, a framework evaluation tool enables data from these assessments to be combined and viewed as a whole picture of national gaps and capacities to inform future strategy. Though use of the framework evaluation tool does not allow identification of a causative relationship between any one preparedness measure or combination of measures and a country's success with preventing in-country transmission, the tool does provide a standardized way of parsing out the relative importance of various activities in contributing to successful response strategies.

This framework evaluation of Uganda's preparedness and response activities related to the 2019 EVD importation events found that preparedness and response activities related to

**Table 3. Framework evaluation for second importation event.**

| August 2019—Importation timeline | Coordination | Surveillance & case finding | Community engagement, risk communication & social mobilization | Health facility preparation | Case referral & management | Lab testing & specimen transport | Vaccination | Refugee-related measures | Logistics & resource mobilization | Point of entry screening | Safe & dignified burials |
|---|---|---|---|---|---|---|---|---|---|---|---|
| **Border crossing** | ✓ | N/a | ✓ | N/a | N/a | N/a | ? | ? | ✓ | ✓ | N/a |
| **Transfer to Bwera ETU** | ✓ | N/a | N/a | ✓ | ✓ | N/a | ? | ? | ✓ | N/a | N/a |
| **Isolation and treatment at Bwera ETU** | ? | N/a | N/a | ✓ | N/a | N/a | ? | ? | ✓ | N/a | N/a |
| **Lab testing** | ✓ | N/a | N/a | N/a | N/a | ✓ | ? | ? | ✓ | N/a | N/a |
| **Remains repatriated** | ✓ | N/a | N/a | N/a | N/a | N/a | ? | ? | ✓ | N/a | N/a |
| **Follow up contact tracing and case finding** | ? | ? | ? | ? | N/a | ? | N/a | ? | ? | ? | N/a |
| **Vaccination activities** | ? | N/a | ? | ? | N/a | N/a | ? | ? | ? | N/a | N/a |
| **Event evaluation** | ✓ | ? | ✓ | ✓ | ✓ | ✓ | ? | ? | ✓ | ✓ | **N/a** |

✓ refers to mechanisms that operated successfully and played a positive contributing role to the response. X refers to mechanisms that operated unsuccessfully or did not play a positive contributing role to the response.? refers to mechanisms that operated with mixed success or limited information; there was uncertainty as to the positive or negative contributing role to the response. N/a refers to mechanisms that were not applicable to response operations.

**Table 4. Summary of framework evaluation results.**

| | Importation Event 1 | Importation Event 2 |
|---|---|---|
| Coordination | ✓ | ✓ |
| Surveillance & case finding | ✓ | ? |
| Community engagement, risk communication & social mobilization | ? | ✓ |
| Health facility preparation | ✓ | ✓ |
| Case referral & management | ✓ | ✓ |
| Lab testing & specimen transport | ✓ | ✓ |
| Vaccination | ? | ? |
| Refugee-related measures | ? | ? |
| Logistics & resource mobilization | ✓ | ✓ |
| Point of entry screening | X | ✓ |
| Safe & dignified burials | ✓ | N/a |

✓ refers to mechanisms that operated successfully and played a positive contributing role to the response. X refers to mechanisms that operated unsuccessfully or did not play a positive contributing role to the response. ? refers to mechanisms that operated with mixed success or limited information; there was uncertainty as to the positive or negative contributing role to the response. N/a refers to mechanisms that were not applicable to response operations.

coordination, health facility preparation, case referral and management, laboratory testing and specimen transport, logistics and resource mobilization, and safe and dignified burials were positive contributors to the response. Other preparedness activities, such as activities related to surveillance, community engagement, vaccination and refugee-related measures, had an unclear role, while preparedness activities related to point of entry screening were indicated to have operated with mixed success according to grey literature and key informant interviews.

It is beyond the scope of the framework evaluation tool developed for this study to determine comprehensively how and why certain preparedness activities that facilitated Uganda's response were successful or unsuccessful, but future analyses with greater amounts of contextual information should explore these questions. For example, when discussing the mixed success of point of entry screening during the June—August 2019 importation events, interviewees noted a number of operational obstacles and considerations necessary for application of the mechanism, some of which have been echoed in peer-reviewed literature [13,14]. The porous nature and massive population movement inherent to the Ugandan border posed a major challenge that could debatably be addressed by strategic planning of screening points at high volume crossings, but it is unknown if this strategic planning or lack thereof was the reason behind point of entry screening being successful for one importation event and not successful for the other without further information. Similarly, preparedness and response activities related to coordination were repeatedly cited in the data as being an integral component of Uganda's successful response, particularly in regards to the importance of cross-border collaboration with neighboring DRC and clear organizational structure and leadership, but it is not clear how coordination at the microplanning or implementation level may have contributed to Uganda's success in preventing in-country transmission.

Other preparedness activity types in the preparedness framework may have performed with mixed or poor success in the framework evaluation, not because they did not play a part in the prevention or interruption of transmission, but because they do not always clearly manifest as a response mechanism that prevents or interrupts transmission. For example, the importance of community engagement, risk communication and social mobilization to facilitate

community buy-in for preparedness measures and response mechanisms cannot be underemphasized, as underscored by both existing best practices in outbreak preparedness and response and key informant interview content. However, the presence and effect of community buy-in directly caused by these activities on response to an importation event is difficult to comprehensively measure and document as opposed to when these measures are not operating successfully and there is an absence of community buy-in (e.g., community distrust). Similarly, it is easier to passively point to the failure of surveillance or vaccination activities rather than when they are successful. Missed cases and missed vaccinations are more obvious and easy to identify than proving and demonstrating that every chain of transmission was found and interrupted or that a vaccinated individual would have become infected if not vaccinated. Additionally, early identification of imported cases by certain mechanisms may have mitigated the need for other preparedness activity types to come into play, which may lead to the potential success of those unused preparedness activity types to go unrecorded since they were not utilized.

Countries responding to similar importation events in the future should consider use of the framework evaluation tool to further validate use of the tool and determine if areas of preparedness that are positive contributors to response activities are consistent or differ by event or national context. Additionally, future uses of the framework evaluation tool should consider expanding use of the tool beyond just EVD importations to include responses to other importation events, such as other viral hemorrhagic fever importations or non-viral hemorrhagic fever importation events. Expanding this framework evaluation tool beyond EVD may highlight or prioritize different preparedness activities. Furthermore, countries that utilize results of the tool to adjust preparedness strategy should share their experiences and lessons learned in order to improve future translations of the tool into policy.

It should be noted that there are various limitations to this analysis. Data from the small number of interviews conducted and the grey literature review did not yield a comprehensive summary of all preparedness activities conducted in Uganda for EVD, but a limited outline or snapshot of activities that occurred. For the grey literature search, limited reports were considered and the majority of those utilized were generated jointly by the WHO country office in Uganda and Uganda's Ministry of Health. Like all grey literature reports, biased authorship or poor validation of reports is a potential issue. In addition, reports were limited with some months having almost daily reports and other months having no reports at all. Other reports appeared to be missing from the overall collection as evidenced by skipping numbered reports. These missing reports may have led to omissions of preparedness and response activities that occurred. Earlier reports provided by WHO Uganda and the Uganda Ministry of Health often had an overall lack of specificity, and therefore, nuances within the response may not have been effectively captured. Key informant interviews were also limited to only a small selection of national-level responders, which may have excluded valuable perspectives at the district- or local-level of response activities. Finally, determination of which activities had positive contributions to preparedness and response to EVD relied upon self-report in the grey literature and by key informants. The absence of objective measures of positive contributions does not exclude the potential for bias.

## Conclusions

Data from grey literature and key informant interviews were utilized to conduct a framework evaluation of preparedness activities conducted in Uganda during the 2018–2020 eastern DRC outbreak. Results of the framework evaluation indicate that preparedness activities focused on coordination, health facility preparation, case referral and management, lab testing and

specimen transport, logistics and resource mobilization, and safe and dignified burials most likely contributed to Uganda's success in preventing sustained in-country transmission during the June—August 2019 EVD importation events although other activities may have also contributed to this success as well. While these findings are not comprehensive or definitive, they may point to areas where countries can invest in EVD preparedness effectively. Future outbreak responses can also utilize similar framework evaluation methods to identify and evaluate areas of investment for preparedness for other outbreaks.

## Supporting information

**S1 Text. Grey literature references.** Please, see the list below for the full list of grey literature (143 documents in total) included to inform the framework evaluation tool in no particular order. All grey literature can be found publicly available online or can be made available upon reasonable request to the corresponding author.
(DOCX)

**S2 Text. Interview guide.** Please, see semi-structured interview guide utilized for key informant interviews.
(DOCX)

**S3 Text Inclusivity in global research questionnaire See responses to journal questionnaire.**
(DOCX)

## Author Contributions

**Conceptualization:** Christina Potter, Lucia Mullen, Steven Ssendagire, Jennifer B. Nuzzo.

**Data curation:** Christina Potter, Lucia Mullen.

**Formal analysis:** Christina Potter, Lucia Mullen.

**Investigation:** Christina Potter, Lucia Mullen, Steven Ssendagire.

**Methodology:** Christina Potter, Lucia Mullen, Steven Ssendagire, Jennifer B. Nuzzo.

**Project administration:** Christina Potter, Lucia Mullen, Steven Ssendagire, Jennifer B. Nuzzo.

**Supervision:** Lucia Mullen, Steven Ssendagire, Jennifer B. Nuzzo.

**Visualization:** Christina Potter.

**Writing – original draft:** Christina Potter.

**Writing – review & editing:** Christina Potter, Lucia Mullen, Steven Ssendagire, Rhoda K. Wanyenze, Alex Riolexus Ario, Doreen Tuhebwe, Susan Babirye, Rebecca Nuwematsiko, Jennifer B. Nuzzo.

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
