## [Decision Letter · Decision Letter 0]

6 Apr 2022

Retrospective identification of key activities in Uganda’s preparedness measures related to the 2018-2020 EVD outbreak in eastern DRC utilizing a framework evaluation tool

PGPH-D-21-00923

Dear Dr. Potter, 

We are pleased to inform you that your manuscript 'Retrospective identification of key activities in Uganda’s preparedness measures related to the 2018-2020 EVD outbreak in eastern DRC utilizing a framework evaluation tool' has been provisionally accepted for publication in PLOS Global Public Health.

Best regards,

Reuben Kiggundu

Academic Editor

Reviewer Comments (if any, and for reference):

Reviewer's Responses to Questions

**Comments to the Author**

1. Does this manuscript meet PLOS Global Public Health’s publication criteria? Is the manuscript technically sound, and do the data support the conclusions? The manuscript must describe methodologically and ethically rigorous research with conclusions that are appropriately drawn based on the data presented.

Reviewer #1: Partly

2. Has the statistical analysis been performed appropriately and rigorously?

Reviewer #1: N/A

3. Have the authors made all data underlying the findings in their manuscript fully available (please refer to the Data Availability Statement at the start of the manuscript PDF file)?

Reviewer #1: Yes

4. Is the manuscript presented in an intelligible fashion and written in standard English?

Reviewer #1: Yes

5. Review Comments to the Author

Reviewer #1: Although that this is not a study researched with primary data (which is one of the key criteria for publishing with PLOS Global Health), it is one worthy of publication because it provides a useful communicable disease outbreak evaluation framework for countries to beef up Disease outbreak emergence preparedness activities.

This paper claims to systematically evaluate specific preparedness activities, capacities and strategies that were most useful in Uganda's containment of EVD in the 2018-2020 outbreak and to the best of my knowledge, the methodology presented and data provided indeed support this claim.

It is amazing to note that all necessary ethical approvals were gotten and all study review limitations noted to foster better reproducibility by other researchers.

6. PLOS authors have the option to publish the peer review history of their article (what does this mean?). If published, this will include your full peer review and any attached files.

**Do you want your identity to be public for this peer review?** For information about this choice, including consent withdrawal, please see our Privacy Policy.

Reviewer #1: **Yes: **Dr. Ifeoluwapo Asekun-Olarinmoye
